# Consumer Response to Food Corporate Social Irresponsibility: Food Performance and Company Ethics Irresponsibility

**DOI:** 10.3390/bs12110461

**Published:** 2022-11-19

**Authors:** Weiping Yu, Dongyang Si, Jun Zhou

**Affiliations:** The Business School, Sichuan University, Chengdu 610065, China

**Keywords:** food performance irresponsibility, company ethics irresponsibility, negative word of mouth, boycott, moral emotion, gender

## Abstract

Corporate social irresponsibility (CSI) seriously damages the rights and interests of stakeholders, particularly consumers. This study analyzes the consumer response to food performance irresponsibility and food corporate ethics irresponsibility by moral emotions. A situational simulation experiment was conducted with the following results: (1) Food performance irresponsibility has the greatest impact on consumer boycotts, while corporate ethics irresponsibility more often leads to consumers’ negative word of mouth (NWOM). (2) Moral emotions play a strong mediating role between CSI and consumers’ NWOM and boycott behavior. (3) Gender significantly moderates the propagation path from moral emotions to NWOM, and female consumers react more strongly to food performance irresponsibility. In conclusion, the paper offers empirical evidence of the effect food corporate social irresponsibility has on consumers’ different responses. Furthermore, it can help food enterprises to identify different CSI types and develop corresponding governance strategies.

## 1. Introduction

Accelerating a new development concept and pattern was fully implemented in China recently. The food industry plays a pivotal role in the economic and social development of this new strategic layout. According to the “China Food Industry Economic Operation Report” statistics in 2021, the food industry accounted for 5.9% of the assets of national industry, created 8.1% of the operating income, and was responsible for 8.5% of the total profit growth. The food industry is, therefore, an important driving force of China’s economic growth. Although the Chinese government has always attached great importance to the social responsibility of the food industry, such as rewarding companies for good performance, the illegal and unethical behavior of food enterprises still continually emerges. Thus, understanding the food corporate social irresponsibility is very essential for food company management and government regulation.

From the perspective of potential injuries to consumers, negative brand events can be divided into performance-related, such as the Shenzhen Wanghong tea shop chain illegally adding the food additive “sunset yellow”, or value-related [1,2], such as Luckin coffee’s accounting fraud, which damaged the shareholders’ rights and interests, or RT-Mart and Carrefour retailers inflating their prices during the pandemic. These events not only have a direct impact on owners but also seriously damage the rights and interests of all shareholders, investors, consumers, and other stakeholders [3,4]. As a core industry, to ensure people’s livelihoods, the food industry not only needs to effectively ensure product quality and safety but should also follow higher moral and ethical standards and actively assume social responsibilities. In particular, food quality is an important tool of competitiveness, thanks to which the consumer perceives brand image [5]. In this study, corporate social irresponsibility (CSI) is defined as “corporate actions that result in (potential) disadvantages and/or harm to other actors” [6]. Thus, from the product and ethical perspectives, we divide food corporate social irresponsibility into two categories: corporate food performance irresponsibility and corporate ethics irresponsibility. Corporate food performance irresponsibility refers to specific food attribute defects or injuries caused by technical errors, standard deviations, and other objective factors or accidental operational errors that lead consumers to doubt whether the brand can meet their functional requirements. Corporate ethics irresponsibility refers to labor accidents caused by neglecting safety and health issues, corporate crimes related to social or ethical issues, and instances of management misconduct, which do not affect the use of specific product attributes and functional products [7]. However, previous studies have not investigated the difference in consumer responses to corporate food performance irresponsibility and corporate ethics irresponsibility.

Emotion is an extension of the field of morality, which refers to an emotional experience produced by individuals when they evaluate their own or others’ behaviors and thoughts according to certain moral standards [8]. CSI behaviors can be seen as moral transgressions, which arise because the corporate wrongdoer is seen to violate the freedom or human dignity of these individuals. Psychologists have shown that when an enterprise violates moral standards because of its misconduct, a series of specific “moral emotions” are generated [9]. Irresponsible corporate behavior can prompt negative emotions such as anger, contempt, and disgust among consumers [10], which leads consumers to exhibit adverse behavior toward enterprises [11]. Scholars of researching consumer behavior are actively establishing an association between the CSI and the responses of consumers [12,13]. In particular, when a company violates environmental regulations or undergoes a product harm crisis, consumers have different reactions. However, most studies have focused on establishing the unconditional direct effects of irresponsible behavior on corporate evaluations and have paid little attention to the underlying moral emotion mechanisms [7]. At the same time, demographic factors, such as age or gender, are usually crucial in all areas of consumer behavior [14]. Scholars note that significant differences exist in information-processing methods between male and female consumers [15]. According to the empathizing systemizing theory, men’s psychology and behaviors are more influenced by cognition, whereas women are more affected by emotion. Accordingly, gender may have moderating effects on the influence exerted by CSI on consumer responses, owing to these differences.

In summary, there are abundant studies on the impact of CSI on consumer responses, but there is a lack of research on food enterprises. Is there a difference in consumer responses between the corporate food performance irresponsibility and corporate ethics irresponsibility? This is worth studying. In addition, few studies have explored the emotional processes and gender regulations behind consumers’ responses to CSI. The current study attempts to address this issue by exploring the moral psychological mechanisms underlying consumer reactions to CSI actions. For the survival and prosperity of food companies, this study uses an experimental method to explore the emotional process behind the response of the CSI of food enterprises to consumers’ negative behavior and psychological mechanisms.

The remainder of this paper is structured as follows: In Section 2, we review the literature and discuss the research hypotheses. In Section 3, we explain the research design and methodology, followed by a report of the results in Section 4. Lastly, in Section 5, we conclude by discussing the reasons for these results, as well as the theoretical contributions, managerial implications, limitations, and future research directions.

## 2. Literature Review and Research Hypotheses

### 2.1. Corporate Social Irresponsibility

Pearce and Manz defined CSI as unethical executive behavior that shows disregard for the welfare of others [16]. It is manifested when executives seek personal gain at the expense of employees, shareholders, other organization stakeholders, and even society at large. Lin-Hi and Müller defined CSI incidents as “corporate actions that result in (potential) disadvantages and/or harm to other actors” [8]. Previous research has used CSI to designate such phenomena, which are characterized by unethical behavior that inflicts harm at different levels of intensity, from death to material loss, for both internal and external corporate stakeholders [17]. CSI can be defined as a violation of the social contract between society and the corporate world. According to social contract theory [18], an important aspect of corporate duty is to avoid causing harm (physical, financial, or mental) to other societal members, including consumers and employees. The conceptualization of CSI further incorporates firm behaviors that incur harmful effects on related or unrelated entities [19,20,21]. The instrumental stakeholder theory of social responsibility holds that a firm’s social negativities are likely to undermine its relationships with stakeholders and, thus, negatively affect shareholders’ assessments [22]. Typical CSI examples include the corporate act of causing physical harm, such as consumers being injured by a product defect [23,24], or a corporate oil spill that contaminates the environment [13,23]. Robson et al. examined vulnerabilities in beef farms. Fake products were the most common type of fraud in the beef industry [25]. Of the reported supply chain incidents, 36.4% were attributed to primary processing, of which 95.5% were counterfeit cases. Furthermore, some CSI examples involve mental harm aspects, such as employee stress caused by employer spying [26] or gender discrimination in the workplace [27].

### 2.2. Negative Moral Reactions toward CSI

Discussions have also emerged on how to address consumers’ negative responses to CSI events [28]. Research suggests that their reactions often happen—in an intuitive way—through spontaneous emotional responses [29]. Automatic emotional reactions were proposed by Haidt and his colleagues in the intuitionist approach to moral behavior [10,30]. Previous studies suggest that bad corporate practices evoke negative moral emotions in consumers. For instance, contempt and anger have been studied as automatic emotional responses to CSI actions in consumer studies [28,31,32]. Both of these overall emotional reactions (i.e., moral emotions) by consumers are proposed to mediate the impact of perceived CSI actions on consumers’ behavioral responses toward the company. 

First, we suggest that a firm’s transgressions induce the negative moral emotion of anger. CSI actions, such as a firm infringing on the rights or freedom of its employees and public stakeholders by the smuggling of imported food, for instance, disadvantage and/or cause harm to other actors [33]. Rozin et al. argued that anger would often be triggered by violations of autonomy codes, as it was often said to be an insult or rights violation [34]. Therefore, righteous anger is the appropriate response to such injuries by people who could be considered victimized. Next, we study a firm’s irresponsibility behaviors, failing to fulfill the firm’s duties and obligations toward society, the economy, and the environment, such as supporting local business partners, protecting the rights and benefits of other members, and contributing to the development and preservation of the environment. Clearly, such actions violate the ethical code, which elicits the negative moral emotion of contempt in people who perceive such CSI incidences. Contempt is often connected to hierarchy, and it is a vertical dimension of social evaluation. It usually manifests as a negative appraisal of others and their actions. Miller argues that contempt stems from the perception that another person does not measure up to either the position they hold or the level of prestige they claim [35]. Similarly, Ekman viewed contempt as disapproving of—and feeling morally superior to—someone. Accordingly, we argue that the corporate community transgressions described above will elicit contempt in consumers [36].

### 2.3. Hypothesis Development

#### 2.3.1. Food Performance Irresponsibility and Company Ethics Irresponsibility in Terms of Consumer Response

CSI has led to increasingly serious negative impacts on companies [23,31]. According to Bulling and Knight, companies, which are considered to be responsible for incidents, have violated corporate commitments, making consumers feel betrayed [37]. A common response is to punish irresponsible companies through boycotts and NWOM [28,38]. At present, many scholars have found that CSI behavior has a negative impact on aspects of consumer response, such as consumer satisfaction, loyalty, and NWOM. Antonetti and Maklan showed that CSI behavior increased the likelihood of consumers being exposed to companies [23]. Malliaris and Urrutia, among others, verified that brand reputation, external response, and corporate response have been evaluated by empirical research on the response of consumers to the brand in crisis [39]. 

Zhuang and Yu further divided the events into product performance, corporate ethical negative events, and compounds of product performance and ethics [40]. Based on this research, Li and Jing considered both the product performance and the product–ethical compound type to belong to enterprises that provide defective products and cause harm to consumers [41]. Based on the literature, the CSI behaviors of food companies are divided into two categories: food performance irresponsibility and company ethics irresponsibility. Food performance irresponsibility is prominent, and it includes companies providing defective food that harms consumers. Company ethics irresponsibility includes the pollution of the environment, damage to the interests of employees and shareholders, and other actions that violate social ethics. Following Lazarus’ appraisal theory, we expect that “being personally affected” will increase the likelihood of boycott participation [42]. Proximity (as a summary construct of being personally, socially, and spatially affected) increases the desire to boycott. Closeness may also be defined in terms of closeness to the product category. Given that proximity is the very cause of boycott motivation, proximity determines the intensity with which consumers search for rationalizations. Thus, proximity has an impact on the intensity of the other drivers of boycott decisions [43]. From the perspective of consumers, food performance irresponsibility is harmful to consumer health, and it is closer than corporate ethics irresponsibility, so the impact on boycott probability is more significant.

Persistently repeating cycles of irresponsible management decisions can cause damage to third parties (e.g., factory workers or environmental damage). Skowronski and Carlston believed that negative information on ethics and morality was more diagnostic than negative information on products/services [44]. Attributes such as management, employees, corporate culture, and negative exposure events for the company’s morality cause the negative perception of external attributes. The present study considers NWOM as a further form of so-called active rebellion, which is regarded as a subtype of consumer resistance. Consumer NWOM can be classified as a prosocial behavior, so idealists consider consumer resistance to be a viable means to remedy human suffering. Therefore, consumers can be altruistically motivated to participate in prosocial acts of resistance. They could engage in NWOM through information sharing to avoid causing harm to others [45,46]. More specifically, NWOM can be triggered by certain motives, including punishing the company through various forms of consumer retaliation [47] and restoring fairness and social justice to serve the larger common good [26,27].

**H1a:** 
*In contrast to corporate ethics irresponsibility, when consumers perceive food performance irresponsibility, the likelihood of a boycott against the company is greater.*


**H1b:** 
*In contrast to food performance irresponsibility, when consumers perceive food corporate ethics irresponsibility, the likelihood of NWOM for the company is greater.*


#### 2.3.2. Mediating Effect of Consumers’ Moral Emotions

Grappi et al. considered corporate social and moral violations to trigger the negative emotions of contempt, anger, and disgust in consumers [28]. Xie et al. found that the irresponsible behavior of companies toward the environment can also cause the above three negative emotions in consumers [48]. Klein and Dawar proved that, in consumers’ perception of CSI behavior, the corporate perception of intentional behavior was higher than unintentional behavior. Consumers expressed more negative emotions, such as anger and hatred [49,50]. The more negative the emotions held by consumers toward CSI behavior, the greater the consumers’ perception value of CSI behavior. Richards et al. found that corporate illegal community and self-governing ethical behaviors trigger the negative moral emotions of anger and contempt, respectively [51].

Boycotts are viewed as a method of emotional expression [43] where negative consumer emotions play a key role in increasing boycott participation [26]. Drawing from the prior literature, consumer boycott motivations can be classified as instrumental vs. non-instrumental [38], with the possibility of mixed motivations for boycott decisions influenced by both [52]. Non-instrumental motivations drive consumers to engage in boycott behaviors based on psychological utility gain or loss. By venting their frustrations, consumers can diminish their negative psychological states and, as a result, experience relief. Unethical corporate behavior may elicit negative emotional reactions, such as consumer outrage. In particular, corporate behavior that arouses consumer outrage relates to the moral domain and has significant societal consequences. Environmental pollution, the toleration of human rights abuses, support for authoritarian regimes, and the exploitation of labor are all important categories of business practices that inspire outrage. For example, buying a children’s toy that consists of hazardous materials may result in dissatisfaction because the toy is not safe, as consumers expect. Consumers may feel outrage because the toy manufacturer committed a moral wrong. Zhao et al. revealed that outrage partially mediated the effects of affective response and disconfirmation on intention to engage in boycott communication [53]. Negative emotions, such as anger and contempt, are important parts of anti-consumption frameworks. In the service recovery literature, emotions have been shown to play crucial mediating roles between consumer perceptions of firm injustice and post-purchase behavioral reactions. This is especially true for negative emotions such as disgust, which constitute a key factor for a better understanding of boycott motivations and behaviors [54]. Lazarus et al. considered behaviors caused by contempt toward rejection and the avoidance of contact, whereas behaviors caused by anger are offensive [42]. Negative emotions, such as contempt, tend to increase people’s willingness to punish offenders, as well as stop or reduce violations. Grappi et al., among others, explored the mediating role of contempt between CSI behavior and consumer response [28]. Scholars such as Xie and Bagozzi have verified the mediating role of contempt in violating community ethics, business ethics, NWOM, and consumer resistance [48]. Therefore, this study explores the mediating role of negative moral emotions (anger and contempt) on the CSI behavior to consumer boycotts of food companies.

**H2a:** 
*Anger plays a mediating role in the relationship between CSI and consumer boycott.*


**H2b:** 
*Contempt plays a mediating role in the relationship between CSI and consumer boycott.*


NWOM is regarded as a pro-social behavior in the literature [38]. Anger may trigger communication among consumers with the aim of convincing other consumers to restore fairness and justice [15]. This may be because NWOM represents a form of individual voice behavior that allows direct personal contact. Andersch [46] also found that consumers perceive NWOM to be particularly suitable for expressing anger and feelings of frustration, as well as benefiting from moral self-enhancement. Apparently, as Verhagen et al. showed, individuals regard NWOM as a means of instantly venting one’s own anger and contempt [55,56]. To this end, the following hypotheses are proposed.

**H3a:** 
*Anger plays a mediating role in the relationship between CSI and NWOM.*


**H3b:** 
*Contempt plays a mediating role in the relationship between CSI and NWOM.*


#### 2.3.3. Moderating Effect of Consumer Gender

Gender is a highly influential variable in consumer behavior, and previous studies have observed gender differences in information processing and behavioral responses. From a psychological perspective, the theory of empathy and systematization holds that there are significant gender differences. Women’s psychology and behavior have historically been considered to be guided by emotion, while men are guided by cognition [57]. A meta-analysis by Chaplin and Aldao suggested that differences in emotional expression between genders are small but significant, with women showing more externalized emotions [58]. George et al. also considered women to be more likely than men to blame companies for product damage crises [59]. Studies have found that female consumers are more intolerant of product failure and show more negative emotions.

This study also investigates gender differences. Women’s outrage may be more pronounced than men’s because consumer outrage is a moral emotion linked to the welfare of other people [10]. These gender differences in moral orientation may affect the process of consumer outrage formation as well, with women potentially being less likely to seek a possible justification for unethical corporate practices. Hence, with respect to consumer outrage and boycott intentions, affective processes may be more important for women. Conformity with gender stereotypes may help to explain these behavioral patterns [60,61]. The results of Lindenmeier also suggested that women’s outrage and behavioral intent were more heavily driven by affective constructs [26]. Thus, women’s outrage is more critical to businesses’ moral transgressions. Moreover, women show contempt for unethical behavioral and environmental cues, resulting in a stronger willingness to draw consequences [62] and to blame companies in the case of corporate ethical misconduct [63]. Bradley et al. revealed that women displayed more extreme reactions, in terms of fear and contempt, to aversive pictures and words [64,65]. Switching to more ethical products is considered a possibility for women in order to appease their conscience and avoid negative feelings [46].

**H4a:** 
*The mediating effect of anger on consumer NWOM was moderated by consumer gender.*


**H4b:** 
*The mediating effect of anger on consumer boycott was moderated by consumer gender.*


**H5a:** 
*The mediating effect of contempt on consumer NWOM was moderated by consumer gender.*


**H5b:** 
*The mediating effect of contempt on consumer boycott was moderated by consumer gender.*


Therefore, this study uses the demographic characteristic of consumer gender as a moderator variable to carry out research. Based on the above analysis of existing research and the inferred hypotheses of this study, we construct a mechanistic model of consumer response to food corporate social irresponsibility behavior, as shown in Figure 1.

## 3. Methodology

### 3.1. Research Design and Measurement

A situational simulation was conducted in this study. The data obtained from this experiment were analyzed using SPSS 22.0. We designed a questionnaire describing two situations related to corporate behavior to measure consumers’ perceptions of CSI behavior and negative moral emotions. Confirmatory factor analysis was used to test the reliability and validity of the measurements. Before regression analysis, multiple collinearity tests were conducted on each variable. Moreover, two-way analysis of variance and independent sample t-tests were used to test consumers’ reactions to different CSI behaviors.

#### 3.1.1. Stimulating Material Design

The purpose of the experiment conducted in this study is to examine consumer boycott and NWOM in response to CSI, the mediating effect of consumers’ negative moral emotions, and the moderating effect of gender. The experiment was conducted using a scenario-based experimental design in the research. Through the effective control of the situation, the experimental method was more effective in controlling external variables, preventing external interference, and obtaining more accurate data. Due to the diachronic investigation of the research object, a change in a variable over a period of time can be obtained. The experimental method is more convincing when proving cause and effect.

The subjects were randomly divided into three groups: two experimental groups and one control group. Participants in the two experimental groups first read a neutral introduction to the virtual dairy product company. Then, they saw news about the virtual dairy product company concerning a product-harm crisis or news about their squeezing of the workforce, while the control group was candidates who received only neutral news about virtual dairy companies. Additionally, after reading the material, participants completed a questionnaire. Participants in the experiment were randomly assigned to one of the three experimental groups.

A dairy product company in China was used in this research. The company’s main business is the development and sale of various dairy products, as well as the breeding and cultivation of dairy cows. At present, the company’s products include fresh milk, lactic acid bacteria drinks, milk powder, and cheese, and it has more than 10,000 employees. In recent years, the company has continuously improved its competitiveness and there has been no large-scale negative news.

In terms of food performance irresponsibility, recently, the company was exposed to food safety issues. After consumers bought brand milk and drank it, they found that the drink did not taste like milk but, instead, like a chemical medicine. According to an investigation by the Quality Supervision Bureau, the equipment of the company had malfunctioned, causing the alkaline detergent used for equipment cleaning to be mixed into the milk. In terms of corporate ethical irresponsibility, the company recently forced employees to work overtime and did not pay them.

The negative behaviors of the food companies in the stimulus materials are CSI behaviors that appear in real life. In the process of designing the stimulus materials, the author invited two doctors to participate in the design, modification, and improvement of the content. Before the formal experiment, a pre-experiment was conducted to test whether the subjects could recognize food performance irresponsibility and corporate ethics irresponsibility. A total of 60 subjects participated in the pre-test (20 in each group). The manipulability question was: “How much do you think the behavior of company A in the above materials conforms to the food performance irresponsibility (company ethics irresponsibility)? 1 = strongly disagree, 7 = strongly agree.” The pre-test results show that both sets of experiments were successfully manipulated.

#### 3.1.2. Measure

The scales used in this study were mature measurement scales. The scale of moral emotions is derived from the research of Grappi et al. [28] and other scholars, including the two emotions of anger and contempt. Each emotion was measured using three items. NWOM includes three items. For boycott behavior, we used the research of Xie et al. [11] with only one item. All variables were measured using a seven-level Likert-style answer sheet, as shown in Table 1.

### 3.2. Data Collection and Sample Composition

The data were gathered from December 2019 to January 2020 using an online questionnaire survey of social media users. We employed a simple random sampling approach and disseminated the online questionnaire on various social media platforms (e.g., WeChat and Weibo). We posted invitations in each questionnaire, including reminders, emphasizing the scientific and non-commercial scope of our investigation. A total of 456 questionnaires were collected, of which 55 invalid questionnaires, with a short response time and the same score for all items, were deleted. Eventually, 401 valid questionnaires were obtained, 134 for each of the two experimental groups and 133 for the control group, with an effective rate of 87.93%. The experimental groups usually exhibited a significant effect for 30 participants. The sample size used met these requirements. The data distribution for each group is presented in Table 2. In the sample, there were more women than men. In terms of age, the predominant age bracket was 26–35 years. In terms of academic qualifications, most participants had a junior college, undergraduate, or postgraduate degree. From an overall point of view, the subjects in this survey were relatively young and well-educated, which met the research requirements.

#### 3.2.1. Reliability and Validity Analysis

We first used Cronbach’s α coefficient to determine the consistency of each variable in the questionnaire for the various items. Except for the low confidence of the contempt sentiment in the food performance group (0.725), all were within the acceptable level. The Cronbach’s α coefficients were all over 0.8, indicating that the various scales in the questionnaire have good reliability. The reliability of each scale used in this questionnaire was tested using the principal component analysis method in the exploratory factor analysis. The results show that the factor loading coefficients of all items were greater than 0.7, indicating that the scale has good validity. The reliability and validity test results of the scale are presented in Table 3.

SPSS 22.0 was used for multiple regression and hierarchical regression to test the hypothesis. To ensure rigor, demographic factors such as the gender, age, and monthly income of the respondents were taken as control variables. Before regression analysis, multiple collinearity tests were conducted on each variable. As shown in Table 4, the results show that the variance inflation factor (VIF) values of each model were <10, and the tolerance was >0.1, which indicates that there was no multiple collinearity problem, and regression analysis could be conducted. 

#### 3.2.2. Manipulation Test

Manipulation inspection tests the manipulative effect of food performance and corporate ethics irresponsibility. Regarding food performance irresponsibility, a score lower than 4 is a judgment error. Among participants, 130 people judged correctly and 4 judged incorrectly, with an error rate of 3.1%. For corporate ethics irresponsibility, a score lower than 4 is also a judgment error. Additionally, 130 people judged correctly and 4 judged incorrectly, with an error rate of 3.1%. The verification operation was, thus, considered successful.

## 4. Results

### 4.1. The Impact of Food Corporate Social Irresponsibility on Consumer Response

Using manipulative issues as independent variables, consumers’ NWOM and boycotts as dependent variables, and demographic variables such as gender, age, and educational background as control variables, regression analysis was performed.

According to the results, both food performance and corporate ethical irresponsibility have a significant impact on consumers’ NWOM and boycotts. Food performance irresponsibility has a significant impact on consumers’ boycotts (B = 0.801, *p* < 0.001) and NWOM (B = 0.348, *p* < 0.001). Corporate ethical irresponsibility behavior has a significant impact on boycott behavior (B = 0.845, *p* < 0.001) and NWOM (B = 0.801, *p* < 0.001), as shown in Table 5. The results support H1a and H1b. According to Table 5, food performance irresponsibility has the strongest impact on stimulating consumer boycotts, and corporate ethics irresponsibility can more strongly cause consumers to engage in NWOM than food performance.

### 4.2. The Mediating Role of Anger and Contempt

This study examines the mediating role of negative emotions between food corporate social irresponsibility and consumer response. By selecting Model 4 and a sample size of 5000, the nonparametric percentile sampling method, with bias correction, was adopted to test the mediating effect of anger and contempt with 95% confidence. CSI behaviors were used as independent variables (food performance irresponsibility being 0 and corporate ethics irresponsibility being 1), anger and contempt as intermediary variables, and boycott and NWOM as dependent variables, alongside demographic variables such as gender, age, and educational background. The mediating effect of the response to CSI, in terms of consumers’ moral emotions and behaviors, is shown in Table 6. The results show that the mediating effect of anger between CSI and consumer boycott is significant (LLCI = 0.0037, ULCI = 0.2523, excluding 0), with a value of 0.1228. The mediating effect of contempt between CSI and consumer boycott is significant (LLCI = 0.0688, ULCI = 0.3488, excluding 0), with a value of 0.2019. The mediating effect of anger between CSI and consumer NWOM is also significant (LLCI = 0.0033, ULCI = 0.2112, excluding 0), with a value of 0.1002. The mediating effect contempt between CSI and consumer NWOM is significant (LLCI = 0.0433, ULCI = 0.2471, excluding 0), with a value of 0.1286. In summary, hypotheses H2a, H2b, H3a, and H3b are supported. Negative moral emotions have a strong mediating effect on the relationship between food corporate social irresponsibility and consumer response.

### 4.3. Moderating Role of Gender

Gender may mediate the effects of moral emotions on consumer responses. Model 14 was selected with gender as the adjusted variable. First, in the food performance irresponsibility group, gender had no moderating effect on the moral emotion of consumers’ boycotts, with a 95% confidence interval that included zero. Regarding the effect of moral emotion in the path from anger to NWOM, a significant moderating effect was demonstrated for women (LLCI = 0.2121, ULCI = 0.5587, excluding 0) but not men. On the path from contempt to NWOM, significant moderation was demonstrated for women (LLCI = 0.0915, ULCI = 0.4198, without 0), while for men, this was less significant. Therefore, the support for H4a and H5a was verified, and H4b and H5b support was rejected. In the corporate ethics group, it can be seen that gender has no moderating effect, as shown in Table 7. This shows that female consumers are more intolerant of food performance irresponsibility, in particular, than male consumers.

## 5. Conclusions

This study investigates food performance irresponsibility, corporate ethics irresponsibility on consumer response combined with negative emotion, and the moderation of gender in the mediation of moral emotion. Our research verifies the following hypotheses, as shown in Table 8.

### 5.1. Discussion

(1) Both food performance and corporate ethics irresponsibility have a significant impact on consumer boycotts and NWOM. Food performance irresponsibility, especially, has a greater impact on consumer boycotts, and corporate ethics irresponsibility has a greater impact on consumers’ NWOM. This is because the different types of CSI affect consumers’ moral judgment to subjectively understand and evaluate the irresponsible behavior of enterprises [66]. Once consumers find that the food has defects, and that it could harm their rights and interests, they adopt more aggressive ways to retaliate against the company. When consumers find that the CSI is related to corporate ethics, they are psychologically alienated. They will demonstrate strong ethical perceptions [67] by communicating negative corporate information to the outside world [68]. This also negatively affects the reputation and image of the business. Accordingly, hypotheses H1a and H1b were supported. (2) Consumers respond to food corporate social irresponsibility through anger and contempt, where contempt plays a more mediating role than anger. CSI behaviors could evoke negative emotions of consumers, and such emotional experiences can further lead consumers to act in ways that harm firms, such as boycotts or spreading negative information [31]. Furthermore, if consumers believe that CSI infringes on ones’ rights, they will become angry. Consumers will also look down on the company even more when the company does not meet the basic institutional norms. Thus, hypotheses H2a, H2b, H3a, and H3b were observed to be supported by the experimental data. (3) The results show that gender moderates the path from negative emotions to consumers’ NWOM in the food corporate social irresponsibility group, but it could not moderate consumers’ boycotts in the two CSI groups. Consumers’ negative emotions will strongly affect their boycott of CSI behavior; as a result, gender may make no difference in the mechanism of moral emotions. Therefore, gender has no moderating effect, and H4b and H5b were rejected. Women’s psychology and behavior are more receptive to emotion than men’s, so negative emotions are more likely to persuade women. Food performance irresponsibility behaviors, especially, are perceived to be more intolerable by female consumers, and they would further share this negative information than men. Thus, hypothesis H4a and H5a were supported. 

### 5.2. Implications

Our study makes important theoretical contributions to the existing literature on the CSI of food companies and customer responses.

Firstly, our finding enriches the research on the CSI of food companies. Previous CSI research has not focused on the nature of the industry, and it has not emphasized the particularity of the food industry. This study highlights the importance of food safety and provides theoretical support for food companies to adopt precise governance strategies for different CSI types. Secondly, from the perspective of products and companies, the differences in consumers’ responses to food performance irresponsibility and corporate ethics irresponsibility are studied. Previous studies have mainly focused on consumers’ moral emotions in response to overall CSI [69], and they have not found that consumers respond to different types of CSI. This expands research into consumer behavior in the field of food corporate social responsibility. Thirdly, this study explores how gender regulates the emotional path from the CSI type to consumer response, laying a theoretical foundation for the consumer response to negative events and new marketing ideas. Previous research has addressed the mediation of emotion between CSI and consumer responses [70], but it ignored the effect of gender on the mediation of moral emotion. In particular, the emotions of female consumers are significantly stronger than those of male consumers, which complements the boundaries of the moral emotion model of CSI.

The findings of this study yield managerial implications for food business practitioners.

First, food companies should be inclined to prevent the occurrence of CSI actions in the first place, particularly food performance irresponsibility. For example, although Luckin Coffee was embroiled in financial fraud, there were no defects with the product itself. Therefore, consumers still chose to buy the product. However, consumers responded to a food safety incident at Burger King by boycotting. We recommend that food companies use emerging technologies, such as the Internet of Things or mobile internet, to ensure food safety of the whole supply chain. Second, food companies should pay attention to change in consumer sentiment in a timely fashion and establish long-term consumer relationships. Our findings show that people have stronger emotional reactions toward corporate wrongdoings. Once such negative incidents occur, companies need to pay close attention to handle both negative emotional reactions and negative attitudes from the public, and they need to take measures to eliminate these emotions to avoid consumers’ negative behaviors [71]. Monitoring emotional consumer reactions toward a company’s practices can be an essential early warning sign of perceived corporate malfeasance and can help firms to respond to problems before they get out of hand. Thus, firms committed to preventing corporate social responsibility (CSR) failures and managing CSR crises can more easily build and strengthen long-term relationships with consumers, as well as contribute to long-term profitability and value creation. Third, after CSI occurs for a food company, different remedial strategies should be adopted according to the gender of the target customer. This study found that gender has a significant regulation impact in the food performance irresponsibility group. It is helpful for food enterprises to reduce the negative impact of CSI through the division of target customer groups. For instance, upon public exposure to irresponsible corporate actions, managers should identify female consumers to adopt crisis response strategies for reducing their subsequent negative responses toward the company. 

### 5.3. Limitations and Future Research

Although this research process was rigorous and scientific, owing to limitations in terms of time and resources, there are some shortcomings. In the questionnaire, the sample was represented by young people with high homogeneity, lacking representability. Although the internal validity of the experimental results was improved, the external validity was reduced. In addition, this study adopted two variables, food performance and corporate ethics, and it did not consider other variables, such as technology, supply chain location, and psychological distance. For example, further research should explore the role of guilt and other negative emotions in brand forgiveness processes [69]. More variables and a larger sample size, including demographic and regional differences, are suggested for further research to facilitate the investigation of causal mechanisms.

## Figures and Tables

**Figure 1 behavsci-12-00461-f001:**
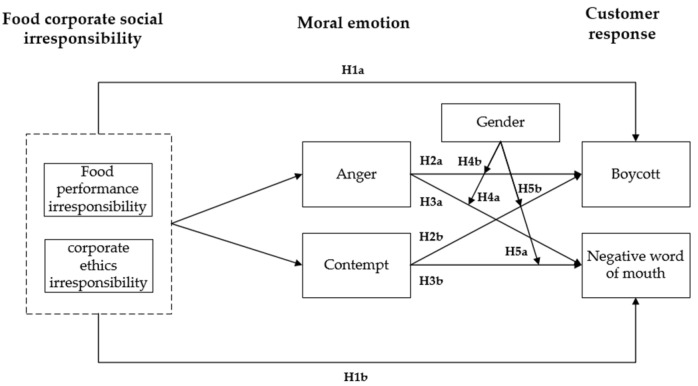
Theoretical model of consumer response to food corporate social irresponsibility.

**Table 1 behavsci-12-00461-t001:** Variable measurement scale.

Variable	Measurement Scale	Source
Anger	Angry	[11,31]
	Mad	
	Very annoyed	
Contempt	Contemptuous	
	Scornful	
	Disdainful	
Negative word of mouth	I intend to say negative things about this company to friends, relatives, and other people.	[28]
	I intend to advise my friends, relatives, and other people not to consider working for this company.	
	I intend to discredit the company to friends, relatives, and other people.	
Boycott	I would put pressure on this company to be socially responsible and correct its bad practices.	[11]

**Table 2 behavsci-12-00461-t002:** Demographic characteristics.

Characteristics	Classification	Rate
Gender	Female	64.6%
	Male	35.4%
Age	Under 25	13.43%
	26 to 35	50.37%
	36 to 45	27.99%
	46 to 55	8.21%
	Above 56	0%
Education level	High school and below	2.99%
	Specialist	13.06%
	Undergraduate	51.87%
	Postgraduate	32.09%
	PhD and above	0

**Table 3 behavsci-12-00461-t003:** Factor loading and reliability of each variable and item.

Variable		Items	Factor Loadings	Reliability	Factor Loadings	Reliability
			**Food Performance**		**Company Ethics**	
Moral emotions	A	A1	0.830	0.832	0.855	0.816
		A2	0.879		0.864	
		A3	0.887		0.847	
	C	C1	0.725	0.726	0.841	0.834
		C2	0.848		0.865	
		C3	0.837		0.893	
Negative word of mouth	N	N1	0.927	0.926	0.930	0.919
		N2	0.956		0.939	
		N3	0.923		0.914	
Boycott		B1				

**Table 4 behavsci-12-00461-t004:** The variance inflation factor.

Model		Unstandardized Coefficient		Standardized Coefficient	*t*	Significance	Collinearity Statistics	
		B	Standard Error	Beta			Tolerance	VIF
1 ^a^	(Constant)	2.525	0.394		6.412	0.000		
	CSI	−0.650	0.090	−0.350	−7.233	0.000	0.926	1.080
	Angry	0.324	0.069	0.296	4.702	0.000	0.547	1.827
	Contempt	0.398	0.063	0.400	6.339	0.000	0.544	1.838
	Gender	−0.125	0.091	−0.065	−1.370	0.172	0.970	1.031
	Age	−0.062	0.053	−0.054	−1.158	0.248	0.986	1.015
	Edu	−0.043	0.059	−0.035	−0.725	0.469	0.939	1.065
2 ^b^	(Constant)	3.330	0.474		7.027	0.000		
	CSI	−0.098	0.108	−0.052	−0.910	0.364	0.926	1.080
	Angry	0.385	0.083	0.348	4.644	0.000	0.547	1.827
	Contempt	0.138	0.076	0.138	1.830	0.068	0.544	1.838
	Gender	−0.086	0.110	−0.044	−0.780	0.436	0.970	1.031
	Age	−0.014	0.064	−0.012	−0.214	0.831	0.986	1.015
	Edu	−0.132	0.071	−0.106	−1.857	0.064	0.939	1.065

a. Dependent variable: boycott. b. Dependent variable: NWOM.

**Table 5 behavsci-12-00461-t005:** Regression coefficient and t value of each main effect test.

**Dependent Variable**	**Y1: Boycott**		**Y2: NWOM**	
	B	*p*	B	*p*
X1: Food performance	0.974	0.000	0.603	0.000
Gender	−0.0279	0.060	0.050	0.307
Age	0.010	0.498	0.077	0.114
Education level	−0.008	0.594	−0.064	0.192
Dependent variable	Y2: Boycott		Y3: NWOM	
	B	*p*	B	*p*
X2: Corporate ethics	0.493	0.000	0.708	0.000
Gender	0.057	0.291	−0.0130	0.773
Age	0.030	0.575	−0.032	0.476
Education level	−0.024	0.659	0.115	0.011

**Table 6 behavsci-12-00461-t006:** The mediating role of negative emotions in terms of CSI and consumer response.

X	Y	M	Effect	*p*	95% Confidence Interval		Does it Contain 0	Significant
CSI	Boycott	Anger	0.1228	0.0635	0.0037	0.2524	No	Yes
		Contempt	0.2019	0.0710	0.0688	0.3488	No	Yes
	NWOM	Anger	0.1002	0.0528	0.0033	0.2112	No	Yes
		Contempt	0.1286	0.0512	0.0433	0.22471	No	Yes

**Table 7 behavsci-12-00461-t007:** The moderating role of gender in terms of CSI, negative moral emotions, and consumer response.

X	Y	M	V		95% Confidence Interval		Does It Contain 0	Significant
Food performance irresponsibility	Boycott	Anger	Direct effect		0.9149	1.0333	No	Yes
			Moderating effect		−0.0489	0.0617	Yes	No
		Contempt	Direct effect		0.9190	1.0362	No	Yes
			Moderating effect		−0.0675	0.0444	Yes	No
	NWOM	Anger	Direct effect		−0.0291	0.4049	Yes	No
			Moderating effect		−0.4283	−0.0474	No	Yes
			Gender	Female	0.2121	0.5578	No	Yes
				Male	−0.0125	0.3149	Yes	No
		Contempt	Direct effect		0.1105	0.5645	No	Yes
			Moderating effect		−0.4452	−0.0213	No	Yes
			Gender	Female	0.0915	0.4198	No	Yes
				Male	−0.1510	0.2248	Yes	No
Corporate ethics irresponsibility	Boycott	Anger	Direct effect		0.7053	0.9825	Yes	Yes
			Moderating effect		−0.613	0.1744	No	No
		Contempt	Direct effect		0.6147	0.8924	Yes	Yes
			Moderating effect		−0.1306	0.2051	No	No
	NWOM	Anger	Direct effect		0.3835	0.8103	Yes	Yes
			Moderating effect		−0.2268	0.1376	No	No
		Contempt	Direct effect		0.3435	0.7886	Yes	Yes
			Moderating effect		−0.2815	0.1181	No	No

**Table 8 behavsci-12-00461-t008:** Hypothesis verification summary.

Research Hypothesis	Verification
H1a: In contrast to corporate ethics irresponsibility, when consumers perceive the food performance irresponsibility, the likelihood of a boycott against the company is greater.	Accepted
H1b: In contrast to food performance irresponsibility, when consumers perceive the food corporate ethics irresponsibility, the likelihood of NWOM for the company is greater.	Accepted
H2a: Anger plays a mediating role in the relationship between CSI and consumer boycotts.	Accepted
H2b: Contempt plays a mediating role in the relationship between CSI and consumer boycotts.	Accepted
H3a: Anger plays a mediating role in the relationship between CSI and NWOM.H3b: Contempt plays a mediating role in the relationship between CSI and NWOM.	Accepted
	Accepted
H4a: The mediating effect of anger on consumer NWOM was moderated by consumer gender.	Accepted
H4b: The mediating effect of anger on consumer boycott was moderated by consumer gender.	Rejected
H5a: The mediating effect of contempt on consumer NWOM was moderated by consumer gender.	Accepted
H5b: The mediating effect of contempt on consumer boycott was moderated by consumer gender.	Rejected

## Data Availability

The dataset of this study is available from the corresponding author on reasonable request.

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
