# Peer review of "Consumer Response to Food Corporate Social Irresponsibility: Food Performance and Company Ethics Irresponsibility"

_behavsci, 2022, doi:10.3390/bs12110461_

Round 1

Reviewer 1 Report

Consumer response to food corporate social irresponsibility: Food performance and company ethics irresponsibility

Dear authors,

Thank you for submitting your paper. This paper has the potential to influence future research streams on a very interesting topic. Still, I have suggestions for several issues I discovered in your review paper.

Abstract

Line 10-12 what u want to communicate is not clear “This study analyzes the mechanism of food performance and corporate ethics irresponsibility in the food industry on negative word of mouth (NWOM) and boycott behavior of consumers, and demonstrates the mediating role of moral emotions and regulating role of gender”. Kindly rewrite these sentences to make them clear.

Line 20-22 the meaning is not clear. Kindly re-phrase.

The abstract should present the takeaway, and what is important. It should include a brief introduction, methodology, analysis, and major findings followed by recommendations. Kindly see the abstract writing style below mentioned articles.

Introduction

Line 28:  what do you mean by CPC?

Line no 37-39: no clear meaning is coming out.

The introduction must conclude with a paragraph summarizing the content of the remaining sections of the paper. (Theoretical Framework, Methodology, Results, Discussion, Conclusions, etc).

Line no 123: the use of sentences like “We propose that bad corporate….” This is not the correct way to do a literature review. Kindly rephrase the sentence.

Line no 119-133: kindly re-write. Things are getting messed up

Line no 133: why the author has suggestions in the review of literature? Kindly make a separate section for the suggestions in the conclusion section or in the discussion section.

Line no 155: “The consumer response path” why there is a change of paragraph?

Line no 199: I don’t agree with the language used for defining the hypothesis. It needs to be improved.

Line no 337: https://doi.org/10.1108/978-1-80262-605-620221004

Line no 104: https://doi.org/10.1016/j.emj.2022.09.012

Line no 254: there is some serious English language issue.

Line no 364: define “ordinary consumers”

What the sampling technique adapted is not clear.

How the questionnaire was distributed online needs to be explained.

The response rate of 87.93% is very high. Site reference for such a high response rate for survey research.

The implications section needs to be included.

Limitations and future research directions are missing.

I suggest you make a table for hypotheses accepted and rejected.

Overall

Check for wording and grammatical flaws. Proofreading is recommended for the article to improve its English. Review the general language of all the text and the common thread.

Review the writing style and use of punctuation marks, since very short sentences are evident within a paragraph and bounded by full stops, and that do not develop a complete idea. This makes it difficult to read the text and goes against the flow and common thread of the text.

Each part of the paper should contribute to the manuscript that is coherent and cohesive. There are several incidences throughout the paper that do not serve this purpose.

Author Response

Dear Reviewer:

Thank you for your comments concerning our manuscript. Those comments you gave are valuable and very helpful in revising and improving our paper. Those also have an important guiding significance to our research. According to your comments, we readjust the relevant contents, and some mistakes and shortcomings are removed so as to meet with approval.

Revision file is presented under the “Track Changes” version. The main corrections in the paper and the responds to the reviewer’s comments are marked in red as following.

  1. Line 10-12 what u want to communicate is not clear “This study analyzes the mechanism of food performance and corporate ethics irresponsibility in the food industry on negative word of mouth (NWOM) and boycott behavior of consumers, and demonstrates the mediating role of moral emotions and regulating role of gender”. Kindly rewrite these sentences to make them clear.

Response: Thank you very much for your suggestion, which is very helpful to optimize the abstract. Line 10-12 “This study analyzes the mechanism of food performance and corporate ethics irresponsibility in the food industry on negative word of mouth (NWOM) and boycott behavior of consumers, and demonstrates the mediating role of moral emotions and regulating role of gender” has been changed to This study analyzes the consumer response to food performance irresponsibility and food corporate ethics irresponsibility by moral emotions.”

  1. Line 20-22 the meaning is not clear. Kindly re-phrase.

Response: We are sorry for unclear meaning in Line 20-22. The conclusion offers empirical research evidence for different types of CSI on consumer responses to ensure food safety, consumer health, and enterprise reputation. Furthermore, this study can help food enterprises identify different consumer responses to CSI types and respond to deficiencies and corresponding governance strategies.has been revised toIn conclusion, the paper offers empirical evidence food corporate social irresponsibility on consumers’ different responses. Furthermore, it can help food enterprises to identify different CSI types and develop corresponding governance strategies.”

  1. The abstract should present the takeaway, and what is important. It should include a brief introduction, methodology, analysis, and major findings followed by recommendations. Kindly see the abstract writing style below mentioned articles.

Response: Thanks the reviewer’s recommended articles, the author has read them carefully. According the suggestions, the abstract has been revised to “Corporate social irresponsibility (CSI) seriously damages the rights and interests of stakeholders, particularly consumers. This study analyzes the consumer response to food performance irresponsibility and food corporate ethics irresponsibility by moral emotions. A situational simulation experiment was conducted, with the following results: (1) Food performance irresponsibility has the greatest impact on consumer boycotts, while corporate ethics irresponsibility more often leads to consumers’ NWOM. (2) Moral emotions play a strong mediating role between CSI and consumers’ NWOM and boycott behavior. (3) Gender significantly moderates the propagation path from moral emotions to NWOM, and female consumers react more strongly to food performance irresponsibility. In conclusion, this study offers empirical evidence food corporate social irresponsibility on consumer different responses. Furthermore, this study can help food enterprises to identify different CSI types and develop corresponding governance strategies.”

  1. Line 28: what do you mean by CPC?

Response: CPC means “Communist Party of China”. The author considers that first sentence is inappropriate and modify to “Accelerating a new development concept and pattern are fully implemented in China recently.”

  1. Line no 37-39: no clear meaning is coming out.

Response: We are sorry that it is not clear meaning in Line no 37-39. “The Chinese government has always attached great importance to the social responsibility of the food industry, including cleaning up the actual status of corporate social responsibility (CSR), and rewarding companies with good performance, etc. Even so, the social irresponsibility such as illegal and unethical behavior of food enterprises still emerge in an endless stream.” was replaced by “Although the Chinese government has always attached great importance to the social responsibility of the food industry, such as, rewarding companies for good performance. The illegal and unethical behavior of food enterprises still continually emerges. So, understanding the food corporate social irresponsibility is very essential for food company management and government regulation.”

  1. The introduction must conclude with a paragraph summarizing the content of the remaining sections of the paper. (Theoretical Framework, Methodology, Results, Discussion, Conclusions, etc).

Response: We have add a paragraph summarizing the content of the remaining sections of the paper in the introduction. The text is “The remainder of this paper is structured as follows: In Section 2, we review the literature and discuss the research hypotheses. In Section 3, we explain the research design and methodology, followed by a report of the results in Section 4. Lastly, in Section 5, we conclude by discussing the reasons for these results, as well as the theoretical contributions, managerial implications, limitations, and future research directions.”

  1. Line no 123: the use of sentences like “We propose that bad corporate….” This is not the correct way to do a literature review. Kindly rephrase the sentence.

Response: “We propose that bad corporate….” has been changed to “Previous studies suggest that bad corporate…”

  1. Line no 119-133: kindly re-write. Things are getting messed up

Response: We have rewritten the Line no119-133. “Discussions have also emerged on how to address consumers’ negative responses to CSI events [28]. ……In this study, we mainly focused on anger and contempt emotions.” has been revised to “Discussions have also emerged on how to address consumers’ negative responses to CSI events [28]. Research suggest that their reactions often happen in an intuitive way- through spontaneous emotional responses [29]. Automatic emotional reactions were proposed by Haidt and his colleagues in the intuitionist approach to moral behavior [10, 30]. Previous studies suggest that bad corporate practices evoke negative moral emotions in consumers. For instance, contempt and anger have been studied as automatic emotional responses to CSI actions in consumer studies [28, 31, 32]. Both of these overall emotional reactions (i.e., moral emotions) by consumers are proposed to mediate the impact of perceived CSI actions on consumers’ behavioral responses to-ward the company.”

  1. Line no 133: why the author has suggestions in the review of literature?

Response: we apologize that there are suggestions in the review of literature. The author has deleted “Hwang, Pan, and Sun found that contempt, anger, and disgust, along with resentment, formed a factor which they termed, media indignation [32], and which mediated the effects of people's exposure to hostile media on their willingness to express criticism of the media, voice their own views, and discuss their opinions with others. In this study, we mainly focused on anger and contempt emotions.”

  1. Kindly make a separate section for the suggestions in the conclusion section or in the discussion section.

Response: The paper has been make managerial implications for food business practitioners in the conclusion section. The text is

“The findings of this study yield managerial implications for food business practitioners.

  • First, food companies should be inclined to prevent the occurrence of CSI actions in the first place, particularly food performance irresponsibility. For example, although Luckin coffee was embroiled in financial fraud, there were no defects with the product itself. Therefore, consumers still chose to buy the product. However, consumers responded to a food safety incident at Burger King by boycotting. We recommend that food companies can use emerging technologies, such as the Internet of Things and mobile internet, to ensure food safety of whole supply chain.
  • Second, food companies should pay attention to changes in consumer sentiment in a timely fashion and establish long-term consumer relationships. Our findings show that people have stronger emotional reactions toward corporate wrongdoings. Once such negative incidents occur, companies need to pay close attention to handling both negative emotional reactions and negative attitudes from the public, and take measures to eliminate these emotions to avoid consumers’ negative behaviors [73]. Monitoring emotional consumer reactions toward a company practices can be an essential early warning sign of perceived corporate malfeasance, and help firms to respond to problems before they get out of hand. Thus, firms committed to preventing CSR failures and managing CSR crises can more easily build and strengthen long-term relationships with consumers, and contribute to long-term profitability and value creation.
  • Third, after CSI occurs for a food company, different remedial strategies should be adopted according to the gender of the target customer. This study found that gender has a significant impact on food performance irresponsibility, which is helpful for food enterprises in reducing the negative impact of CSI on consumers through the division of target customer groups. For instance, upon public exposure to irresponsible corporate actions, managers should identify female consumers to adopt crisis response strategies for reducing their subsequent negative responses toward the company. ”
  1. Line no 155: “The consumer response path” why there is a change of paragraph?

Response: “The consumer response path” is followed by the hypothesis, so there is a change of paragraph. And, “The consumer response path” has been revised to “Hypothesis Development”.

  1. Line no 199: I don’t agree with the language used for defining the hypothesis. It needs to be improved.

Response: We have improved the language used for defining the hypothesis H1 and H2. “H1: In contrast to corporate ethics irresponsibility, when consumers perceive the CSI behavior of food performance irresponsibility, the stronger the boycott against the company. H2: In contrast to food performance irresponsibility, when consumers perceive the CSI behavior of food corporate ethics irresponsibility, the higher the NWOM for the company.” has changed to “ H1: In contrast to corporate ethics irresponsibility, when consumers perceive food performance irresponsibility, the likelihood of a boycott against the company is greater. H2: In contrast to food performance irresponsibility, when consumers perceive food corporate ethics irresponsibility, the likelihood of NWOM for the company is greater.”

  1. 1 Line no 337: https://doi.org/10.1108/978-1-80262-605-620221004

Line no 104: https://doi.org/10.1016/j.emj.2022.09.012

Response: The author has read these paper carefully, and borrowed the abstract writing style of these articles.

  1. Line no 254: there is some serious English language issue.

Response: The author has checked for wording and grammatical in the whole paper. “Research regards NWOM as a pro-social behavior [38], which is regarded as a subtype of consumer resistance. Anger motivates people to restore fairness and justice [55] to achieve this goal. Anger may trigger communication among consumers and try to convince other consumers.” has been revised to “NWOM is regarded as a pro-social behavior in the literature [38]. Anger may trigger communication among consumers with the aim of convincing other consumers to restore fairness and justice [55].”

  1. Line no 364: define “ordinary consumers”

Response: Ordinary consumers means “The person who purchases products or services from a person or business frequently”, actually, consumers.

  1. What the sampling technique adapted is not clear.

Response: We employed a simple random sampling approach.

  1. How the questionnaire was distributed online needs to be explained.

Response: We disseminated the online questionnaire on various social media (e.g., WeChat and Weibo).

  1. The response rate of 87.93% is very high. Site reference for such a high response rate for survey research.

Response: The data was gathered using online questionnaire survey of social media, which is has a very high transmission. So, the response rate is also high.

  1. The implications section needs to be included.

Response: We have make implications section in conclusion. The text is

“Our study makes important theoretical contributions to the existing literature on the CSI of food companies and customer responses.

  • Our study enriches the research on the CSI of food companies. Previous CSI research has not focused on the nature of the industry, emphasizing the particularity of the food industry. This study highlights the importance of food safety, and provides theoretical support for food companies to adopt precise governance strategies for different CSI types.
  • From the perspective of products and companies, the differences in consumers’ responses to food performance irresponsibility and corporate ethics irresponsibility are studied. Previous studies have mainly focused on consumers’ moral emotions in response to overall CSI [71], and have not found that consumers respond to different types of CSI. This expands research into consumer behavior in the field of food corporate social responsibility.
  • This study explored how gender regulates the emotional path from the CSI type to consumer response, laying a theoretical foundation for the consumer response to negative events and new marketing ideas. Previous research has addressed the mediation of emotion between CSI and consumer responses [72], but ignored the effect of gender on the mediation of moral emotion. In particular, the emotions of female consumers are significantly stronger than those of male consumers, which complements the boundaries of the moral emotion model of CSI.

The findings of this study yield managerial implications for food business practitioners.

  • First, food companies should be inclined to prevent the occurrence of CSI actions in the first place, particularly food performance irresponsibility. For example, although Luckin coffee was embroiled in financial fraud, there were no defects with the product itself. Therefore, consumers still chose to buy the product. However, consumers responded to a food safety incident at Burger King by boycotting. We recommend that food companies can use emerging technologies, such as the Internet of Things and mobile internet, to ensure food safety of whole supply chain.
  • Second, food companies should pay attention to changes in consumer sentiment in a timely fashion and establish long-term consumer relationships. Our findings show that people have stronger emotional reactions toward corporate wrongdoings. Once such negative incidents occur, companies need to pay close attention to handling both negative emotional reactions and negative attitudes from the public, and take measures to eliminate these emotions to avoid consumers’ negative behaviors [73]. Monitoring emotional consumer reactions toward a company practices can be an essential early warning sign of perceived corporate malfeasance, and help firms to respond to problems before they get out of hand. Thus, firms committed to preventing corporate social responsibility (CSR) failures and managing CSR crises can more easily build and strengthen long-term relationships with consumers, and contribute to long-term profitability and value creation.
  • Third, after CSI occurs for a food company, different remedial strategies should be adopted according to the gender of the target customer. This study found that gender has a significant impact on food performance irresponsibility, which is helpful for food enterprises in reducing the negative impact of CSI on consumers through the division of target customer groups. For instance, upon public exposure to irresponsible corporate actions, managers should identify female consumers to adopt crisis response strategies for reducing their subsequent negative responses toward the company. ”
  1. Limitations and future research directions are missing.

Response: There is limitations and future research directions in conclusion. The text is “Although this research process was rigorous and scientific, owing to limitations in terms of time and resources, there are some shortcomings. In the questionnaire, the sample was represented by young people with high homogeneity, lacking representability. Although the internal validity of the experimental results was improved, the external validity was reduced. In addition, this study adopted two variables, food performance and corporate ethics, and did not consider other variables, such as technology, supply chain location, and psychological distance. For example, further research should explore the role of guilt and other negative emotions in brand forgiveness pro-cesses [71]. More variables and a larger sample size including demographic and regional differences are suggested for further research to facilitate the investigation of causal mechanisms.”

  1. I suggest you make a table for hypotheses accepted and rejected.

Response: The author has add table 9 for hypotheses accepted and rejected.

Table 9. Hypothesis verification summary.

Research hypothesis

Verification

H1a: In contrast to corporate ethics irresponsibility, when consumers perceive the food performance irresponsibility, the likelihood of a boycott against the company is greater.

Accepted

H1b: In contrast to food performance irresponsibility, when consumers perceive the food corporate ethics irresponsibility, the likelihood of NWOM for the company is greater.

Accepted

H2a: Anger plays a mediating role in the relationship between CSI and consumer boycotts.

Accepted

H2b: Contempt plays a mediating role in the relationship between CSI and consumer boycotts.

Accepted

H3a: Anger plays a mediating role in the relationship between CSI and NWOM.

H3b: Contempt plays a mediating role in the relationship between CSI and NWOM.

Accepted

Accepted

H4a: The mediating effect of anger on consumer NWOM was moderated by consumer gender.

Accepted

H4b: The mediating effect of anger on consumer boycott was moderated by consumer gender.

Rejected

H5a: The mediating effect of contempt on consumer NWOM was moderated by consumer gender.

Accepted

H5b: The mediating effect of contempt on consumer boycott was moderated by consumer gender.

Rejected

  1. Check for wording and grammatical flaws. Proofreading is recommended for the article to improve its English. Review the general language of all the text and the common thread.

Response: The author has checked for wording and grammatical in the whole paper.

  1. Review the writing style and use of punctuation marks, since very short sentences are evident within a paragraph and bounded by full stops, and that do not develop a complete idea. This makes it difficult to read the text and goes against the flow and common thread of the text.

Response: The author has checked the writing style and use of punctuation marks in the whole paper.

  1. Each part of the paper should contribute to. There are several incidences throughout the paper that do not serve this purpose.

Response: The author has checked the whole manuscript to keep the article coherent and cohesive.

We tried our best to improve the manuscript and made some changes in the manuscript. These changes will not influence the content and framework of the paper.

We appreciate for your warm work earnestly, and hope that the correction will meet with approval.

Once again, thank you very much for your comments and suggestions.

Dongyang Si

Sichuan University

Reviewer 2 Report

- Do you have ethical approval for this research? If yes, state it in this methodology section

-The conclusion section is written in bullet points; try to remove the number and put it in essay format. 

Author Response

Dear Reviewer:

Thank you for your comments concerning our manuscript. Those comments you gave are valuable and very helpful in revising and improving our paper. Those also have an important guiding significance to our research. According to your comments, we readjust the relevant contents, and some mistakes and shortcomings are removed so as to meet with approval.

Revision file is presented under the “Track Changes” version. The main corrections in the paper and the responds to the reviewer’s comments are marked in red as following.

  1. Do you have ethical approval for this research? If yes, state it in this methodology section

Response: Our research conforms the ethical approval. In the methodology, we have add the text “We posted invitations in each questionnaire, including reminders, emphasizing the scientific and non-commercial scope of our investigation “

  1. The conclusion section is written in bullet points; try to remove the number and put it in essay format.

Response: The author has removed the number,and wrote in bullet points in the conclusion section.

We tried our best to improve the manuscript and made some changes in the manuscript. These changes will not influence the content and framework of the paper.

We appreciate for your warm work earnestly, and hope that the correction will meet with approval.

Once again, thank you very much for your comments and suggestions.

Dongyang Si

Sichuan University

Reviewer 3 Report

I highly appreciate both the theoretical and empirical part of the article. The authors presented the mechanism of food performance and corporate ethics irresponsibility in food industry on negative word of mouth (NWOM) and boycott behavior of consumers, and demonstrates the mediating role of moral emotions and regulating role of gender. The authors constructed a theoretical model of food CSI and consumer response, combined with negative emotion variables, focusing on consumer gender regulation, and explored the differential impact of food performance, corporate ethics, and consumer response.

The article has the correct layout, the literature review is written correctly, the authors refer to the current literature, and properly conducts a scientific discussion.

The research method (situational simulation experiment) is appropriately selected to achieve the research goal and verify research hypotheses. The used statistical methods are not very advanced, but they are sufficient to achieve the research goals.

The authors drew the right conclusions from the research, which are useful not only for scientists, but also for companies in the food sector.

I congratulate the authors the research idea and wish them further high-quality publications.

Author Response

Dear  Reviewer:

Thanks very much for your positive comments. We tried our best to improve the manuscript again for ensuring publication standards. Revision file is presented under the “Track Changes” version.

Once again, thanks very much for your approval.

Dongyang Si

Sichuan University

Round 2

Reviewer 1 Report

Dear authors,

Thank you for submitting your paper. This paper can potentially influence future research streams on an exciting topic. Still, I have suggestions for several issues I discovered in your review paper.

I recommend improving the discussion section. kindly discuss each hypothesis and the related arguments in support and against it. the discussion section can not be in points.

Write conclusions in paragraphs and not in points. Conclusions should reflect your research questions and the results of hypothesis testing.

kindly include the implications sections as well. Each part of the paper should contribute to the manuscript that is coherent and cohesive.

Author Response

Dear reviewer,

Thanks again for your reply. The main corrections in the paper and the responds to your comments are marked in red as following:

Point 1. I recommend improving the discussion section. kindly discuss each hypothesis and the related arguments in support and against it. the discussion section can not be in points.

Response 1: Thank you very much for your suggestion. The discussion section has been shown in paragraphs. And, the author has further added the related arguments in support and against hypotheses. “(1) Both food performance and corporate ethics irresponsibility have a significant impact on consumer boycotts and NWOM… and they would further share these negative in-formation than men.” has revised to “(1) Both food performance and corporate ethics irresponsibility have a significant impact on consumer boycotts and NWOM. Especially, food performance irresponsibility has a greater impact on consumer boycotts, and corporate ethics irresponsibility has a greater impact on consumers’ NWOM. This is because the different types of CSI affect consumers' moral judgment to subjectively understand and evaluate the irresponsible behavior of enterprises [68]. Once consumers find that the food has defects, and that it could harm their rights and interests, they adopt more aggressive ways to retaliate against the company. When consumers find that the CSI is related to corporate ethics, they are psychologically alienated. They will demonstrate strong ethical perceptions [69] by communicating negative corporate information to the outside world [70]. This also negatively affects the reputation and image of the business. Accordingly, hypotheses H1a and H1b were supported. (2) Consumers respond food corporate social irresponsibility through anger and contempt, where contempt plays a more mediating role than anger. CSI behaviors could evoke negative emotions of consumers, and such emotional experiences can further lead consumers to act in ways that harm firms, such as boycott or spreading negative information [31]. Furthermore, consumers believe that CSI infringes on ones’ rights, they will become angry. Consumers will also look down on the company even more, when the company does not meet the basic institutional norms. Thus, hypotheses H2a, H2b, H3a, and H3b were observed to be supported by the experimental data. (3) The results shows that gender moderates the path from negative emotions to consumers’ NWOM in food corporate social irrespon-sibility group, but could not moderate to consumers’ boycott in two CSI group. Consumers’ negative emotions will strongly affect their boycott to CSI behavior, as a result, gender may have no difference in the mechanism of moral emotions. Therefore, gender has no moderating effect, and H4b and H5b were rejected. Because women’s psychology and behavior are more receptive to emotion than man, negative emotions are more likely to persuade women. Especially, food performance irresponsibility behavior are perceived more intolerable by female consumer, and they would further share these negative information than men. So, hypothesis H4a and H5a were supported.”

Point 2. Write conclusions in paragraphs and not in points. Conclusions should reflect your research questions and the results of hypothesis testing.

Response 2: “5. Discussion and conclusion” has been changed to “5. Conclusions”, which includes “5.1. Discussion”,” 5.2. Implications” and “5.3. Limitations and future research”. The text “This study investigates food performance irresponsibility, corporate ethics irresponsibility on consumer response combined with negative emotion, and the moderation of gender in the mediation of moral emotion. Our research verifies the following hypotheses, as shown in Table 9.” has been added in first paragraph of conclusions section to reflect our research questions and the results of hypothesis testing. And, conclusions has written in paragraphs and not in points.

Point 3. kindly include the implications sections as well. Each part of the paper should contribute to the manuscript that is coherent and cohesive.

Response 3: In “5.2. Implications”, we deleted “This research divides the CSI of food companies into food performance and corporate ethics. Both types of CSI have significant effects on consumer responses, and there are obvious differences determined by consumers’ emotions. Consumers can make subjective evaluations of CSI responses according to their own perceptions and are affected by the significant moderating effects of their gender.”, and only retained the text “Our study makes important theoretical contributions to the existing literature on the CSI of food companies and customer responses.” in order to implications more highlighted. The author has added conjunction (firstly, secondly, thirdly) to make implications more coherent and cohesive.

Dongyang Si

Sichuan University